# Telemedicine in Emergency Medicine in the COVID-19 Pandemic—Experiences and Prospects—A Narrative Review

**DOI:** 10.3390/ijerph19138216

**Published:** 2022-07-05

**Authors:** Malgorzata Witkowska-Zimny, Barbara Nieradko-Iwanicka

**Affiliations:** 1Department of Human Anatomy, Medical University of Warsaw, 5 Chalubinskiego Str., 02-004 Warsaw, Poland; 2Chair and Department of Hygiene and Epidemiology, Medical University of Lublin, 7 Chodzki Str., 20-093 Lublin, Poland; barbara.nieradko-iwanicka@umlub.pl

**Keywords:** telehealth, telemedicine, COVID-19, emergency department, emergency medicine

## Abstract

Even before the year 2020, telemedicine has been proven to contribute to the efficacy of healthcare systems, for example in remote locations or in primary care. However, with the outbreak of the COVID-19 pandemic, telehealth solutions have emerged as a key component in patient healthcare delivery and they have been widely used in emergency medicine ever since. The pandemic has led to a growth in the number of telehealth applications and improved quality of already available telemedicine solutions. The implementation of telemedicine, especially in emergency departments (EDs), has helped to prevent the spread of COVID-19 and protect healthcare workers. This narrative review focuses on the most important innovative solutions in emergency care delivery during the COVID-19 pandemic. It outlines main categories of active telehealth use in daily practice of dealing with COVID-19 patients currently, and in the future. Furthermore, it discusses benefits as well as limitations of telemedicine.

## 1. Introduction

The COVID-19 pandemic resulting in a sudden appearance of a large number of patients infected with the severe acute respiratory syndrome coronavirus 2 (SARS-CoV-2) is one of the greatest challenges to healthcare systems worldwide. In the COVID-19 pandemic, healthcare providers needed not only to help numerous patients with a new, highly-contagious disease, but also to provide care to patients with chronic diseases, emergency conditions, or other health conditions and diseases, and not expose them to SARS-CoV-2. 

Despite effective prevention, such as vaccinations, new waves and evidence of new variants of COVID-19 are still expected to emerge. Therefore, the situation should be thoroughly analyzed and conclusions should be drawn from the experiences from the onset and the course of the pandemic for new effective solutions to be developed and implemented. 

One of the results of the COVID-19 pandemic is a rapid adoption of digital tools and technologies such as telemedicine and virtual healthcare in healthcare systems. The terms “telehealth”, “mobile health” (mHealth) and “telemedicine” are often used interchangeably. Telemedicine is defined as providing remote healthcare services using information and communication technologies [1]; however, at its core, it was about supporting long-distance clinical care, professional health-related education, public health, and health administration. Hence, although telehealth (mHealth, eHealth) is often referred to or referenced as telemedicine (which refers specifically to remote clinical services), it embraces a wider variety of healthcare services, including those provided by professionals other than physicians, such as nurses and pharmacists. It may include clinical care services, education for both patients and providers, and public health or healthcare administrative services [2]. Telehealth can be used as a tool to monitor, diagnose, treat, and counsel patients in circumstances where in-person care is not feasible, or when telehealth is more convenient or economical. Although telehealth has been in use for a few decades now, the COVID-19 pandemic made apparent its many advantages and previously unthought of uses. Moreover, the use of telemedicine in the COVID-19 pandemic has been promoted by international organisations, such as, e.g., the World Health Organisation [3]. Numerous studies have investigated the use of telemedicine in Emergency Departments (EDs), and many papers have discussed telehealth methods and attempted to determine the feasibility of these systems for emergency settings [4]. EDs are critical to the management of acute illness and injury, and therefore the provision of healthcare system access. The overcrowding of emergency departments may be a serious public health issue worldwide whereas reducing interpersonal contact has been one of the least expensive and most widely used COVID-19 control strategies. The pandemic and excess patient flow into emergency departments may be a huge challenge for emergency medicine [5]. There is evidence suggesting that the employment of telemedicine accelerates triage, as well as positively impacts both patient care and cost reduction in the healthcare systems [6]. However, not all healthcare providers and patients are equally ready for a wide-scale implementation of telehealth solutions. Structured efforts need to be made to assess the skills demanded from telehealth/telemedicine workers and to amend their education and training. The aim of this narrative review is to show the possible application of telemedicine in pre-hospital, inpatient, and post-emergency departments in the pandemic and beyond. 

## 2. Telehealth Application in Emergency Medicine

Emergency departments have typically used telehealth for remote consultations. However, as the pandemic evolved new areas of the possible use of telehealth services have been identified also in emergency medical systems. One of these new areas is telemedical support for paramedics/ambulance nurses as an important factor in effective prehospital emergency care, especially due to the pandemic-related shift from traditional on-scene physician missions to telemedical-supported missions. Innovative telehealth solutions may be implemented in (i) prehospital/pre-ED settings, (ii) within ICUs and Emergency Departments, (iii) post-ED discharge, and (iv) education.

### 2.1. Prehospital/Pre-ED Telehealth

Telemedicine may enhance the quality of prehospital emergency medical services by improving remote prehospital consultations, helping to dispatch urgent patient transfer, and enhancing the supervision of healthcare providers. The excess flow of patients into emergency departments was a major issue early in the pandemic. Patients with COVID-19 symptoms or under investigation for COVID-19 reported to EDs increasing the risk of spreading the virus to other patients. These experiences demonstrate the need for prehospital and emergency department coordination to best serve communities during unpredictable situations related to the COVID-19 pandemic. The use of telehealth services may help to coordinate emergency systems, and thus improve the functioning of the whole healthcare system. 

Numerous studies in emergency medicine and telehealth have investigated the use of telemedicine in prehospital/pre-ED settings [7,8,9,10]. A significant increase in the number of patients admitted with SARS-CoV-2 infections was found during the early stage of the COVID-19 pandemic, which indicates difficulty in obtaining medical help at the beginning of the pandemic, before the implementation of pre-hospital teleconsultations [9]. Tele-triage decreased waiting time for treatment, and thus reduced the overcrowding resulting from the input factors [10].

A small number of studies describe telemedicine used for a remote evaluation of patients before on-site care with evaluation being conducted not only by the ED staff but also by medical students, nurses, and physicians [11,12]. 

Telemedicine has the potential to improve the quality of prehospital emergency medical services by addressing triage and by collaboration with local and governmental emergency services providers [13]. Pre-hospital telehealth systems have been developed to provide faster, more efficient care for COVID-19 patients triaged to appropriate levels of care. New technologies including mobile units, telemedicine, and wearable technology have been implemented to optimize this process [12].

The main interventions in prehospital/pre-ED settings were to evaluate and screen patients prior to ED presentation and to communicate with patients prior to their arrival in ED. In some countries, additional COVID-19 dedicated call centers, together with additional call takers, even such as volunteers, allowed the main ED dispatchers to focus on routine emergency calls [14,15]. It seems that cooperation with national emergency medical services organizations in some countries, from the beginning of the COVID-19 pandemic, helped them slow down the spread of the disease. This was done by enabling suspected patients to avoid hospitals and community clinics and by managing the treatment of mild cases of COVID-19 at home, allowing hospitals to cater for more severe cases. 

Paramedics in ambulances would often hold telemedicine meetings with specialists to facilitate pre-hospital diagnosis and reduce treatment delays in progressive respiratory failure of COVID-19 patients [16]. Telehealth allowed for better decision-making for boarded ED patients awaiting transfer to Intensive Care Units or other hospitals [17]. Tele-triage, defined as a videoconference between emergency medical services and EDs, is the subject of only one paper [18].

Telemedicine has also been used by emergency medicine doctors to supervise nurse practitioners and general practitioners in long-term care facilities or skilled nursing facilities. The main goal of these interventions was to enable appropriate triage of patients at risk for COVID-19 disease and to compensate for the lack of medical resources [19].

In conclusion, telemedicine may contribute to supporting prehospital decision-making for diagnoses, lifesaving interventions, and hospital destinations. Most importantly, telemedicine can be used to ensure that healthcare resources are used in the most efficient and effective way.

### 2.2. Telehealth within ICUs and Emergency Departments

The following areas of interest can be distinguished in the application of telehealth within the Emergency Departments: (i) saving personal protective equipment (PPE); (ii) assessment and remote monitoring of ED and ICU patients; (iii) interaction of specialist services with ED physicians, or how specialists may interact with patients either for emergency conditions or in the ED; (iv) remote supervision of trainees by attending physicians and (v) enabling patient contact with “virtual visitors”.

Telehealth initiatives and virtual communication decrease patient and ED staff exposure to COVID-19 by limiting personal contact and, thereby, decreasing the use of PPE [20]. To limit the exposure and curb the spread of COVID-19 in EDs, some hospitals eagerly deployed telehealth software used for ED inpatients to communicate with all patients, regardless of whether they were being admitted or discharged. Consultant services and hospitalists were able to contact the patients via their devices and assess them remotely [21,22,23,24]. Moreover, in case of foreign-speaking patients, the healthcare provider could easily invite a certified interpreter to join in the conversation to improve communication. Emergency departments have also used telehealth as a screening tool for acute care needs to limit staff and patient exposure to the virus and the use of PPE. 

In spite of its high cost, there has been an increased interest in remote patient monitoring (RPM) ever since the outbreak of the coronavirus (COVID-19) pandemic [25,26]. The tele-ICU systems (with audio and visual virtual solutions) have been developed to address the increasing demand for intensive care services and the shortage of intensivists. The implementation of tele-ICU offers potential advantages and makes critical care delivery more efficient in disasters or pandemics. Risk prediction algorithms, smart alarm systems, and machine learning tools augment conventional coverage and, as research has shown, can potentially improve the quality of care [27]. Moreover, not only do RPM programs allow for closer monitoring of patients, but also trends in patient reporting may reflect clinical courses of COVID-19 that may otherwise have remained uncharacterized [28]. Steinberg at all., after deploying an RPM programme to monitor and triage patients tested positive for SARS-CoV-2, observed a reduction in ED utilization or even hospital admission and ICU care [29].

Numerous medical centers with no access to professional RPM systems used a combination of nursing phone calls, telemedicine appointments, and other remote communication methods to minimize the spread of the virus. 

Even the best virtual ICU care systems and remote patient monitoring fall short of providing clinical decision support tools required to effectively care for critically ill patients. Hence, specialty consultations are common and important in emergency medical practice. They affect patient flow. A rising number of COVID-19 and non-COVID-19 ED patients increased the need for specialist consultations (internal medicine, anesthesia, radiology, trauma surgery, pulmonary and critical care medicine). Limited human resources and the need for reducing healthcare worker exposure have resulted in a rising number of consultations via telemedicine in many EDs. Thus, in the COVID-19 pandemic, telemedicine was used for no-touch patient evaluations by ED specialists [30]. Conversations with patients regarding discharge planning or follow-up, as well as discussions with health care providers about patient care plans, tele-supervision of low-risk procedures and disposition planning, were held virtually [31].

Due to an increased risk of morbidity and mortality among senior medical doctors, some measures have been taken to decrease the risk of COVID-19 exposure for these healthcare professionals, e.g., by introducing virtual oversight over residents and non-advanced practice providers [32].

Telehealth was also used by emergency physicians infected with COVID-19 who were well enough to work from home, but not yet beyond the recovery period allowing them to return to physical work [33]. Virtual hospitalist programs expanded the ability to confront the challenges of the COVID-19 crisis at the epicenter of the pandemic and expanded mechanisms to train and support new or inexperienced hospitalists to provide and expand health care services. A virtual hospitalist program expanded a mechanism to train and support new or inexperienced hospitalists to provide and expand palliative care services [34]. COVID-19 infections have raised numerous safety concerns. To comply with infection control concerns, access to visitors/family members was restricted for both COVID-19 and non-COVID-19 patients in ICUs. Hence, a new research area emerged for investigating virtual visiting under strict personal visitation policies in hospital wards. Critical care units would substitute in-person visits with virtual visitation programs, e.g., by video chat to ease stress on patients and family members to improve communication [35].

Conclusions regarding the use of telehealth in EDs and ICUs are as follows: ED patients and staff were able to decrease exposure to the virus, save PPE and improve communication between patients and healthcare providers, as well as patients and their families [7,18,35].

### 2.3. Post-ED Discharge Telehealth

In the context of post-ED discharge telehealth, were explored the extension of ED care through remote patient monitoring (RPM) at home or in long-term care facilities [36]. Remote patient monitoring with the transmission of physiological data from the home setting to clinicians, is typically used in the management of chronic conditions such as diabetes or hypertension [37]. Suitable mobile health solutions used for post-ED discharge patient monitoring enable COVID-19 patients to be routinely followed-up outside the hospital setting. In the pandemic, this also reduces potential disease spread and prevents the overloading of healthcare systems. There have been a variety of RPM programs recently discussed in the literature. However, few of these programs were developed to address coronavirus disease patients in particular [38,39]. In a study from New York, discharged patients were provided with pulse oximeters, thermometers, and a symptom reporting app. The data collected by the software may help not only to secure particular patients who may benefit from early intervention but also to trace the natural history of COVID-19, enabling evidence-based identification of the course of infection. Moreover, there are mobile-assisted respiratory rehabilitation programs for COVID-19 convalescents. Some research data indicate the need to develop new education and training programs with a focus on the interdisciplinary rehabilitation of patients with post-COVID-19 syndrome.

The reduction in the availability of inpatient rehabilitation for non-COVID-19 patients, for example post-stroke patients, due to the pandemic, has had many consequences. Telerehabilitation contributes to significant patient improvement, at the same time, it eliminates the problem of transport, which is cited as a major limitation in the access to inpatient therapy [40,41].

Further research is needed to understand the efficacy, cost, risk, and implications of using RPM in the acute or post-acute settings for COVID-19 as well as for other diagnoses.

### 2.4. Tele-Education

The COVID-19 pandemic has affected many aspects of human life and it is clear that it has had a serious impact on medical education by disrupting the traditional education of students, future health care providers, medical staff and their continuing medical education, postgraduate medical education, as well as patient health education [42,43].

Practical training of healthcare personnel has been one of the biggest logistical challenges in the COVID-19 pandemic. Many different teaching and learning strategies had to be implemented, e.g., technology-enhanced learning (TEL), simulation-based learning, technology-based clinical education, mobile learning, and blended learning [44].

Video tutorials with interactive instructions for numerous procedures, e.g., putting on and removing PPE or the use of ventilator service, allowed healthcare workers to develop the necessary skills and perform the procedures step-by-step (COVID-19 and SIM Program Trains for Proper and Efficient Use of PPE) [45]. Research has demonstrated that using in situ simulations to improve the effectiveness of PPE in COVID-19 improved teamwork and the work of individual team members. 

One of the biggest challenges of the COVID pandemic was the shortage of medical staff and the need for newly trained clinicians/health care students, to enter workforce without compromising the integrity of core learning outcomes. Some pilot programs were created in which students remotely participated in COVID-19 ward rounds via videoconferencing. The virtual bedside rounds were implemented to successfully engage students in learning about the diagnosis and treatment of COVID-19 [46,47].

Despite the pandemic, tele-OSCEs (Objective Structured Clinical Examinations) were carried out with appropriate planning, consensus building, and technology readiness assessment. Tele-exams played a critical role in sustaining the flow of health care students into the workforce during the pandemic [48]. 

Distance learning has its merits and is an important tool in medical education. However, medical students and medical staff need to develop clinical, practical, hands-on medical examination skills, and communication skills [49]. Hence, it is impossible for even the best tele-education to replace traditional ways of acquiring practical skills.

On the other hand, tools, methods, and learning resources associated with these distance learning strategies have the potential to improve the learners’ level of knowledge and performance by access to online learning resources such as Massive Open Online Courses, virtual clinical cases, or blended courses. Nonetheless, the long-term impact of these learning methods on medical education remains unknown.

## 3. Telehealth Use Perspectives

Telemedicine holds great promise in facilitating emergency medical practice and, as research shows, it is increasingly being used in emergency medicine. The use of remote monitoring devices capable of obtaining physiological parameters remotely and the creation of a machine learning derived risk score may facilitate triage in outpatients with COVID-19 and other disorders, or in accident situations. All this is particularly suited to medical emergencies where treatment delays adversely affect clinical outcomes. The ambulance nurse/paramedics are often the first point of contact for the patient with healthcare in an emergency situation. The first applications were developed for transmitting real-time video to facilitate consultation between prehospital health care providers and regional medical support (RMS) for ambulance care [50]. Live imaging allows to reach a consensus on the patient’s current medical care needs and contributes to a feeling of increased patient safety in the ambulance. Hence, video consultations are a likely future not only within ICUs and Emergency departments but also in prehospital settings [51,52]. 

Although most EDs had some prior experience with telehealth, the COVID-19 pandemic has accelerated the implementation of telemedicine. The use of telecommunications technology reduced the risk of exposure to the virus, decreased ED visit volume, conserved personal protective equipment, and contributed to human resource optimization. After the pandemic, RPMs will be more frequently used in EDs.

There are several issues to be taken into consideration by individual hospitals, EDs, or emergency management departments that wish to implement telehealth applications bearing in mind possible subsequent waves of COVID-19 or future disasters/pandemics. One of them is offering medical telehelp in multiple languages, which is particularly difficult to implement quickly but is needed in a multinational setting. Technical and infrastructural issues in implementing such software solutions may have an impact on patient safety. Hardware integration is yet another issue. Healthcare facilities should consider investing in a stockpile inventory of relatively low-cost, validated devices that can be repurposed as needed. EDs and hospitals should develop staffing plans, and consider engaging not only retired clinicians, and medical students, but also technical assistance to accommodate for new telehealth programs.

It is important to assess how new technology will interact with the existing protocols and how it will affect load balancing across multiple systems. 

Greater efforts should be made to ensure the availability of all telehealth services, not only those related to combating the effects of SARS-CoV-2 infections. Telehealth has many advantages, especially during an infectious disease outbreak. However, not all healthcare providers and patients are equally ready for a wide-scale implementation of telemedicine. Hence, an obvious change to medical training is required across all specialties. In the post-COVID era, all healthcare providers need to be prepared to provide remote medical services.

Some educational innovations brought about by the COVID-19 pandemic will be maintained post-pandemic. Online learning improves the level of both patients’ and healthcare providers’ knowledge and it may be a valuable supplement to health education and staff continuing medical education. Without any doubt, tele-education and virtual training components will never completely replace hands-on, face-to-face clinical experience, but they may supplement the traditional approaches with more participant-centric and convenient content.

Ongoing research will be needed to assess the factors that may affect such an adaptation by both healthcare providers and patients as well as the quality and clinical outcomes associated with telemedicine implementation in all health fields, including emergency medicine.

## 4. Limitations of Telehealth Use

The COVID-19 pandemic resulted in a major paradigm shift in health care delivery with the universal adoption of telemedicine. Some researchers highlight the crucial role of online tools in promoting public health, especially during the pandemic [53]. However, telehealth has major limitations. Current literature points to some of the barriers associated with, e.g., funding, time, infrastructure, equipment, skills, or preference for a face-to-face consultation. Elderly low-educated or low-income patients with little or no access to computers or smartphones are less likely to use video-enabled telehealth and communication portals. Some ethical issues have also been raised, including autonomy, beneficence, non-maleficence, justice, and professional-patient relationships [54]. 

One of the biggest problems faced by patients and doctors during remote consultations is the lack of a physical examination, which may lead to misdiagnosis or delay in diagnosis [55,56]. Moreover, the lack of direct physical examination reduces the patient’s preferences for virtual consultations in the future [57]. Ramaswamy et al., in a retrospective observational cohort study, compared in-person and video visits in the pre–COVID-19 period to in-person and video visits during the COVID-19 period, respectively. In adjusted analyses, video visits and the COVID-19 period were associated with higher patient satisfaction. Surprisingly, younger age and female gender were associated with lower patient satisfaction [58]. A high level of satisfaction with telehealth was observed in studies across medical specialties, but emergency medicine was not included in any of the research [59]. Lack of opinions of emergency department patients may result from the difficulties in measuring patient satisfaction in a situation of an imminent threat to one’s life. COVID-19 has made health care providers move rapidly from traditional personal appointments to telephone or video consultations in challenging circumstances. Before the COVID-19 pandemic, technological challenges, professional scepticism and ethical, financial, administrative or legal barriers had limited the uptake and use of remote consultations, ensuring they accounted for a limited proportion of patient consultations [58,60]. The overall assessment of mHealth by patients and the medical staff is inconclusive. Some doctors and patients found telemedicine more convenient than in-person consultations, pointing to benefits that included COVID-19 safety, no need to travel, and reduced waiting times as benefits. This was especially true for people who felt comfortable with quick checks, prescriptions or administrative inquiries, or for those patients who had difficulty getting to the appointment. However, some healthcare professionals, especially clinicians, have expressed concern that telemedicine may be misused as a cost- and time-saving but not in the best interest of patients. Publications that would represent the opinion of clinicians have not been found. One of the important factors in favor of a face-to-face meeting is the establishment of a doctor-patient relationship. It is notable that the attitudes of patients to the health systems have changed. The concept of the passive patient is now outdated and has been replaced by the concept of a patient who is more active and involved in all processes. More robust data on eHealth’s long-term efficacy, safety, and costs are necessary.

Health professionals report gaps in knowledge and skills needed for the safe and effective use of digital tools by healthcare professionals. Governments have published guidelines for telehealth practice and explicit patient consent is required for remote consultations to be provided [61]. Principles of medical ethics, including professional standards to protect the privacy and confidentiality of patients, should be binding, observed and complied with. Suitable training, enhanced documentation templates, guidelines for communication and observing regulations for information management may contribute to a decrease in the number of dangers and pitfalls associated with remote consultations.

The integration of telehealth in the curricula for nursing and medical students varies across study programs. The level of medical staff competencies for using telehealth technology to deliver care and manage specific disorders varies as well [62].

Some regulatory barriers and low reimbursement rates are also obstacles to widespread telehealth use. Insurers and policymakers should consider appropriate payment mechanisms, reimbursement rates, and financing for telehealth appointments because telephone or video encounters are similar in length, content, and quality to face-to-face consultations. It should be mentioned at this point that cybersecurity and the protection of personal health information are one of the most frequently overlooked, yet one of the most crucial aspects of telemedicine research with security risks, confidentiality issues, and unauthorized access to medical data emerging as major concerns [63].

A fully-functional application of telehealth solutions in EDs is yet to come, and there are many unknown areas that need to be investigated. There are many factors, both internal and external to the ED which play a role in determining the sustainability of telemedicine in the acute care setting, in the case of trauma, unforeseen emergency events, or managing the pandemic situations. Findings from many pandemic studies provide data-informed insights into possible guidelines for employing telemedicine to increase healthcare system resilience to future health crises.

## 5. Conclusions

The evolution of telemedicine is one of the biggest changes that have occurred in emergency medicine in the COVID-19 pandemic. Advantages and barriers to the adoption of telehealth practices in emergency department services are summarized in Table 1.

The main goal of telemedicine in the COVID-19 pandemic has been to reduce direct face-to-face contact in the hope of curbing virus transmission and healthcare providers’ protection while ensuring high standards of treatment and therapy not only to the SARS-CoV-2 infected but also to all patients. There have been numerous attempts to harness new technologies in prehospital settings, within ICUs and hospitals (such as RPM), and in post-ED discharge emergency medical services to meet the demands placed on the healthcare systems in the COVID-19 pandemic. Despite its many advantages, telehealth does have a number of limitations. However, during the COVID-19 pandemic, patients, healthcare providers, administrators, and policymakers could see that the telemedicine model works. This cannot be undone. In the 21st century, telehealth services will be expected as part of routine and integrated service provision, medical training, and professional activity.

## Figures and Tables

**Table 1 ijerph-19-08216-t001:** Advantages and disadvantages of telemedicine in emergency practice in the COVID-19 pandemic.

Advantages		Disadvantages	
Highlights	Reference	Highlights	Reference
Remote patients’ evaluation	[11,12,21,22,23,24]	Misdiagnosis or delay in diagnosis because of lack of a physical examination	[55,56]
Reduction of the exposure to COVID-19 by limiting personal contact	[7,20,32,33]	Lack of learning of clinical, practical, and hands-on medical skills by medical staff and students	[49]
Triage acceleration	[6,11,12]	Lack of health care providers’ preparation and professional scepticism	[58,60]
Reduction of the overcrowding in EDs	[10]	Lack of patient readiness and low patient satisfaction	[58]
Saving personal protective equipment	[18,20]	No access to digital tools	[54]
Telemedical support of medical caregivers and decision-making processes	[16,17,30,31,50]	Problems with protecting the privacy and confidentiality of patient data	[61]
Fast communication with foreign-speaking patients	[21,22]	Lack of telehealth in the curricula study programs	[62]
Closer and permanent patient monitoring in ICUs and at home.	[25,26,28,29,36,37]	Regulatory, legal, and administrative barriers	[58,60,63]
Better coordination of emergency systems	[45]	Low financing for telehealth appointments	[63]
Supervision of healthcare providers	[34,52]	Huge costs of cybersecurity and the protection of personal health information	[61]
Virtual visitors—substitution of in-person visits by remote contact between family members	[35]	Difficulties in the creation of doctor-patient relationships.	[58]
Continuation of student and health care providers’ medical education and patient health education.	[45,46,47,48]	No possibility of assessing practical medical skills	[44,45,48]

## Data Availability

Not applicable.

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
