# Peer review of "Telemedicine in Emergency Medicine in the COVID-19 Pandemic—Experiences and Prospects—A Narrative Review"

_ijerph, 2022, doi:10.3390/ijerph19138216_

Round 1

Reviewer 1 Report

This is welll written and comprehensive  I think there should be more discussion about patient's and provider response to telemedicine.  

Author Response

Response: Thank you for your nice opinion. We really appreciate your comments which helped us improve the quality of our manuscript. The remark of the Reviewer#1 was included and mark in the text: page 7, line 313-342.

Reviewer 2 Report

Thank you for submitting the manuscript for my review. This work addresses the development of telemedicine during the SARS-CoV2 pandemic. It is shown as a narrative review, the informative value is unfortunately insufficient. The text is difficult to read; I think a graphical or tabular presentation of the key statements of each chapter would be useful. Telemedicine in emergency services existed before the pandemic, which has certainly been pushed. It will be relevant in the future to counteract the lack of specialists and to better serve rural regions. Overall, I don't see any added value to publishing the narrative review.

1)     The title is somewhat confusing. Paragraph 2.3 "Post-ED Discharge Telehealth" does not describe emergency care, but the clinical follow-up afterwards. Paragraph 2.4 "Tele-education" is also not focused on emergency medicine alone. Therefore, I recommend adjusting the title. It is unclear to me if this is a field report or if an unselective literature search was done.

2)     The text is difficult to read because there was no visualization in the form of tables or graphs. I recommend presenting important findings graphically or in tabular form. Advantages and disadvantages of telemedicine could be tabulated. The perspectives of telemedicine should also be presented separately in a table.

For example:

Author

Topic

Source

Prehospital…

Emergency Departments

ICU

Post-ED Discharge Telehealth

3)     Paragraph 2.1 “Numerous studies…” (2 sources)… “A small number…” (2 reference sources)… More sources should be listed under "numerous" after all.

4)     Lines 91-93, 93-96: Please insert reference source.

5)     Lines 258-261: Please insert reference source.

6)     Data protection should also be discussed. Can patients refuse to have a video shot?

7)     Lines 293-295: “a major paradigm shift”…  They write about barriers and "major limitations" and provide no evidence for the paradigm shift. Since telemedicine costs and will cost a lot of money, I currently believe that it will not be pushed further.

8)     Conclusions: “This it cannot be undone” Here again is a question of money to implement telemedicine. It also shows that when the infection numbers fall down, a lot of things go back into old structures. Here, however, the shortage of specialists (for example, emergency physicians) could be discussed as an opportunity for telemedicine.

Author Response

Thank you for submitting the manuscript for my review. This work addresses the development of telemedicine during the SARS-CoV2 pandemic. It is shown as a narrative review, the informative value is unfortunately insufficient. The text is difficult to read; I think a graphical or tabular presentation of the key statements of each chapter would be useful. Telemedicine in emergency services existed before the pandemic, which has certainly been pushed. It will be relevant in the future to counteract the lack of specialists and to better serve rural regions. Overall, I don't see any added value to publishing the narrative review.

Response: We appreciate your comments. Thank you for your opinion, however, it is opposite to the opinion of the other four reviewers.

1)     The title is somewhat confusing. Paragraph 2.3 "Post-ED Discharge Telehealth" does not describe emergency care, but the clinical follow-up afterwards. Paragraph 2.4 "Tele-education" is also not focused on emergency medicine alone. Therefore, I recommend adjusting the title. It is unclear to me if this is a field report or if an unselective literature search was done.

 Response: Thank you for your comments. The aim of our narrative review is to show a broad perspective on telehealth applications in emergency medicine. Work on a meta-analysis on this subject is in progress. According to the suggestions Reviewer#5,  we should include the general context of telehealth not only in emergency medicine alone. In our opinion, there is a balance between the general context and description of telehealth use in emergency medicine.

2)     The text is difficult to read because there was no visualization in the form of tables or graphs. I recommend presenting important findings graphically or in tabular form. Advantages and disadvantages of telemedicine could be tabulated. The perspectives of telemedicine should also be presented separately in a table.

For example:

Author

Topic

Source

Prehospital…

Emergency Departments

ICU

Post-ED Discharge Telehealth

Response: Response: This was corrected per the comment. Please find Table 1.

3)     Paragraph 2.1 “Numerous studies…” (2 sources)… “A small number…” (2 reference sources)… More sources should be listed under "numerous" after all.

Response: This was corrected per the comment.

4)     Lines 91-93, 93-96: Please insert reference source.

Response: This was corrected per the comment.

5)     Lines 258-261: Please insert reference source.

Response: This was corrected per the comment.

6)     Data protection should also be discussed. Can patients refuse to have a video shot?

Response: This was corrected per the comment: page 7, line 344-350.

7)     Lines 293-295: “a major paradigm shift”…  They write about barriers and "major limitations" and provide no evidence for the paradigm shift. Since telemedicine costs and will cost a lot of money, I currently believe that it will not be pushed further.

Response: In our humble opinion, the health care service has already changed in many countries and we can not see huge barriers to stopping the progress of civilization. Of course, we respect the opposite opinion, however, contrary to the reviewer, we believe that it will be still pushed further.  Healthcare technology has leaped to the fore to help healthcare providers manage their patients better by reducing the dangers inherent in personal contact, waiting in crowded waiting rooms or laboratories, and hospitalizations. Applications of mHealth can promote healthy behaviors for primary or secondary disease prevention, help with the self-management of chronic illnesses, improve provider training, and cut down on visits to the doctor. At the same time, they can help personalize interventions to an unprecedented level. Artificial Intelligence (AI) technology and virtual reality have just been present in our lives

8)     Conclusions: “This it cannot be undone” Here again is a question of money to implement telemedicine. It also shows that when the infection numbers fall down, a lot of things go back into old structures. Here, however, the shortage of specialists (for example, emergency physicians) could be discussed as an opportunity for telemedicine.

Response: Thank you for your comment. Please see page 4, line 172-181.

Reviewer 3 Report

This is an incredibly well done review of ED Telehealth during the COVID-19 pandemic. It includes several areas where tele-health was implemented and review the limitations as well. 

Below are some suggestions for the authors:

Section 2.1 Prehospital/pre-ED Telehealth

Line 84--begging-->beginning

I would love to see some discussion in here (if other papers exist) about increasing clinic Telehealth providing an ability for patients to be seen by PMDs and thus not have to use ED Telehealth or come to the ED thus reducing the ED burden

2.2 Telehealth within ICUs and EDs

line 134 translator-->interpreter (translation is written, interpretation is spoken)

2.4 Tele-education

Authors could consider including information on tele-rounds for medical students

https://onlinelibrary.wiley.com/doi/full/10.1111/medu.14512

4 Limitations of Tele-health Uses

I would like for authors to include information on the inability to perform a physical exam via tele-health and how this can result in adverse patient outcomes (delay in diagnosis, incorrect diagnosis, etc)

Author Response

Response: Thank you for your nice opinion. We really appreciate your comments which helped us improve the quality of our manuscript.

Below are some suggestions for the authors:

Section 2.1 Prehospital/pre-ED Telehealth

Line 84--begging-->beginning

Response: This was corrected per the comment.

I would love to see some discussion in here (if other papers exist) about increasing clinic Telehealth providing an ability for patients to be seen by PMDs and thus not have to use ED Telehealth or come to the ED thus reducing the ED burden.

Response: Unfortunately, we did not find such data. Retrospective Cohort Study has done in New York experienced an 8729% increase in video visit utilization during the COVID-19 pandemic compared to the same period last year.

2.2 Telehealth within ICUs and EDs

line 134 translator-->interpreter (translation is written, interpretation is spoken)

Response: This was corrected per the comment.

2.4 Tele-education

Authors could consider including information on tele-rounds for medical students

https://onlinelibrary.wiley.com/doi/full/10.1111/medu.14512

Response: This was corrected per the comment: page 5, line 236-239 and references: 46, 47.

4 Limitations of Tele-health Uses

I would like for authors to include information on the inability to perform a physical exam via tele-health and how this can result in adverse patient outcomes (delay in diagnosis, incorrect diagnosis, etc)

Response: This was corrected per the comment: page 7, line 313-326.

Reviewer 4 Report

Dear editors,

Thank you for the opportunity to do this review. This article describes a narrative review about telemedicine in emergency medicine in the COVID-19 pandemic. I found the article really interesting and informative. I think that the authors have performed a good review, and the text can be helpful for other researchers. Nevertheless, there are some minor aspects that I think could be revised. These are the following:

- I would propose the authors use the term eHealth instead of telemedicine. I understand that the latter is older, but telemedicine, in the beginning, was related to phone consultations. eHealth is more focused on the use of information and communication technologies. The authors describe that «The terms “telehealth”, “mobile health” (mHealth) and “telemedicine” are often used interchangeably». Albeit it is true for many articles, this is not exact, as the terms refer to different aspects of telemedicine. Indeed, they describe that «Telemedicine is defined as providing remote healthcare services using information 39 and communication technologies» (which is correct and a different concept than eHealth or mHealth). Therefore, I would invite them to be more precise with the terms, as this is a review of this interesting topic.

- In «2.2. Telehealth within ICUs and Emergency Departments», the authors write that «Conclusions regarding the use of telehealth in EDs and ICUs are as follows: ED patients and staff were able to decrease exposure to the virus, save PPE and improve communication between patients and healthcare providers, as well as patients and their families». I would propose to add one or more citations of articles that support this important affirmation.

- The use of eHealth has critical social implications. The authors describe some of the advantages and disadvantages of using eHealth. Nevertheless, there are more aspects regarding these aspects of eHealth that could be mentioned.

Author Response

Response: Thank you for your nice opinion. We really appreciate your comments which helped us improve the quality of our manuscript.

Thank you for the opportunity to do this review. This article describes a narrative review about telemedicine in emergency medicine in the COVID-19 pandemic. I found the article really interesting and informative. I think that the authors have performed a good review, and the text can be helpful for other researchers. Nevertheless, there are some minor aspects that I think could be revised. These are the following:

- I would propose the authors use the term eHealth instead of telemedicine. I understand that the latter is older, but telemedicine, in the beginning, was related to phone consultations. eHealth is more focused on the use of information and communication technologies. The authors describe that «The terms “telehealth”, “mobile health” (mHealth) and “telemedicine” are often used interchangeably». Albeit it is true for many articles, this is not exact, as the terms refer to different aspects of telemedicine. Indeed, they describe that «Telemedicine is defined as providing remote healthcare services using information 39 and communication technologies» (which is correct and a different concept than eHealth or mHealth). Therefore, I would invite them to be more precise with the terms, as this is a review of this interesting topic.

Response: This was corrected per the comment: page 1 and 2, line 41-48.

- In «2.2. Telehealth within ICUs and Emergency Departments», the authors write that «Conclusions regarding the use of telehealth in EDs and ICUs are as follows: ED patients and staff were able to decrease exposure to the virus, save PPE and improve communication between patients and healthcare providers, as well as patients and their families». I would propose to add one or more citations of articles that support this important affirmation.

Response: The remark of the Reviewer#4 was included. page 4, line 193.

- The use of eHealth has critical social implications. The authors describe some of the advantages and disadvantages of using eHealth. Nevertheless, there are more aspects regarding these aspects of eHealth that could be mentioned.

The remark of the Reviewer#4 was included. Please see Table 1.

Reviewer 5 Report

The paper presents a narrative review focused on the critical, innovative solutions in emergency care delivery during the COVID-19 pandemic. It outlines the main categories of active telehealth use in the daily practice of dealing with COVID-19 patients currently and in the future.

The paper is well organized, and the length is appropriate. The title is chosen correctly, and the abstract provides sufficient information to give a clear idea of what to expect from the paper.

In general, the technical depth of the paper meets the requirements for a scientific article published in a quality journal.

Major comments:

More articles dealing with Telemedicine in emergency medicine in a general context and not only for COVID-19 should be reviewed in the Introduction section.

All conclusions from each section should be consolidated in the Conclusions section.

Minor comments:

The authors use the words "tele-health" and "telehealth". They should only use one of them.

Author Response

Response: Thank you for your nice opinion. We really appreciate your comments which helped us improve the quality of our manuscript.

Major comments:

More articles dealing with Telemedicine in emergency medicine in a general context and not only for COVID-19 should be reviewed in the Introduction section.

Response: Thank you for your comments. However, according to the suggestions Reviewer#2 and the manuscript's title we should focus on emergency medicine alone. In our opinion, there is a balance between the general context and description of telehealth use in emergency medicine.

All conclusions from each section should be consolidated in the Conclusions section.

Response: According to the suggestion of Reviewer#2 we decided to prepare the table with advantages and disadvantages. Please see Table 1.

Minor comments:

The authors use the words "tele-health" and "telehealth". They should only use one of them.

Response: This was corrected per the comment.

Round 2

Reviewer 2 Report

Thank you for the revision. All the best for you.